# *Photorhabdus luminescens* TccC3 Toxin Targets the Dynamic Population of F-Actin and Impairs Cell Cortex Integrity

**DOI:** 10.3390/ijms23137026

**Published:** 2022-06-24

**Authors:** Songyu Dong, Weili Zheng, Nicholas Pinkerton, Jacob Hansen, Svetlana B. Tikunova, Jonathan P. Davis, Sarah M. Heissler, Elena Kudryashova, Edward H. Egelman, Dmitri S. Kudryashov

**Affiliations:** 1Department of Chemistry and Biochemistry, The Ohio State University, Columbus, OH 43210, USA; dong.630@buckeyemail.osu.edu (S.D.); pinkerton.56@buckeyemail.osu.edu (N.P.); hansen.185@buckeyemail.osu.edu (J.H.); kudryashova.1@osu.edu (E.K.); 2The Ohio State Biochemistry Program, The Ohio State University, Columbus, OH 43210, USA; 3Department of Biochemistry and Molecular Genetics, University of Virginia, Charlottesville, VA 22903, USA; weilizheng1116@gmail.com (W.Z.); egelman@virginia.edu (E.H.E.); 4Department of Physiology and Cell Biology, Davis Heart and Lung Research Institute, The Ohio State University, Columbus, OH 43210, USA; tikunova.1@osu.edu (S.B.T.); davis.812@osu.edu (J.P.D.); sarah.heissler@osumc.edu (S.M.H.)

**Keywords:** TccC3, actin, ADP-ribosylation, bacterial toxin, calponin-homology domain, actin dynamics, Cryo-EM, spontaneous nucleation, blebbing

## Abstract

Due to its essential role in cellular processes, actin is a common target for bacterial toxins. One such toxin, TccC3, is an effector domain of the ABC-toxin produced by entomopathogenic bacteria of *Photorhabdus* spp. Unlike other actin-targeting toxins, TccC3 uniquely ADP-ribosylates actin at Thr-148, resulting in the formation of actin aggregates and inhibition of phagocytosis. It has been shown that the fully modified F-actin is resistant to depolymerization by cofilin and gelsolin, but their effects on partially modified actin were not explored. We found that only F-actin unprotected by tropomyosin is the physiological TccC3 substrate. Yet, ADP-ribosylated G-actin can be produced upon cofilin-accelerated F-actin depolymerization, which was only mildly inhibited in partially modified actin. The affinity of TccC3-ADP-ribosylated G-actin for profilin and thymosin-β4 was weakened moderately but sufficiently to potentiate spontaneous polymerization in their presence. Interestingly, the Arp2/3-mediated nucleation was also potentiated by T148-ADP-ribosylation. Notably, even partially modified actin showed reduced bundling by plastins and α-actinin. In agreement with the role of these and other tandem calponin-homology domain actin organizers in the assembly of the cortical actin network, TccC3 induced intense membrane blebbing in cultured cells. Overall, our data suggest that TccC3 imposes a complex action on the cytoskeleton by affecting F-actin nucleation, recycling, and interaction with actin-binding proteins involved in the integration of actin filaments with each other and cellular elements.

## 1. Introduction

*Photorhabdus luminescens* are motile entomopathogenic bacteria that mutualistically associate with *Heterorhabditis* nematodes [1]. Upon invasion into insect larvae, nematodes kill the insect by releasing *P. luminescens* producing highly potent toxic proteins [2]. Accordingly, *P. luminescens* are widely used in the agricultural industry as a biological insecticide [3]. Although *P. luminescens* primarily target insects, human infections caused by this and other related species of the *Photorhabdus* genus have been reported [4,5,6,7].

One of the major virulent factors produced by *P. luminescens* is the 1.7-MDa Toxin complex (Tc) which consists of TcA, TcB, and TcC components. TcA forms a giant ~1.4-MDa pentameric pre-pore complex that is soluble but transforms into a membrane-integrated pore complex under acidic pH [8,9]. TcB and TcC components complete the formation of the holotoxin by binding to the TcA pentamer. These subunits form a protective cocoon surrounding the effector domain, represented by the C-terminal portion of the TcC subunit [10]. While the structural/delivery components of the holotoxin (i.e., TcA, TcB, and N-terminal TcC) are conserved, the C-terminal domains of TcC are variable and designated as hypervariable regions (hvr). These catalytically active domains are released from TcC by autoproteolysis and, upon translocation via the TcA pore, define the specific toxicity of the entire complex [11]. Due to the high selectivity and efficiency, Tc toxins are considered insecticidal agents for genetically engineered plants [12].

Two of the *P. luminescens* TcC-hvr effectors (TccC3-hvr and TccC5-hvr; hereafter, TccC3 and TccC5, respectively) are mono ADP-ribosyltransferase (mART) toxins, both targeting the actin cytoskeleton. TccC3 ADP-ribosylates actin at threonine-148 (T148), causing extensive clustering of filamentous actin (F-actin) [11]. TccC5 ADP-ribosylates Rho, Rac, and Cdc42 small GTPases at conserved glutamine residues, promoting actin stress fiber formation. As a result, both toxins promote the inhibition of phagocytosis in insect hemocytes [11,13].

While many bacterial mART toxins modify actin at R177, generating non-polymerizable actin species [14], only TccC3 is known to modify actin at T148, engaging very different pathogenicity mechanisms. T148 locates in actin subdomain-3 (SD3) near the hydrophobic cleft, which is recognized as a hot spot for interaction with actin-binding proteins (ABPs) and myosin motors [15]. Several of these ABPs play critical roles in the cytoskeleton organization. Thus, ADP-ribosylation of T148 has been shown to weaken the interaction of globular actin (G-actin) with thymosin-β4 (TMSB4)—a 34-a.a. peptide involved in the maintenance of the non-polymerizable pool of G-actin. For similar reasons, TccC3-ADP-ribosylated-F-actin was found to be resistant to depolymerization by cofilin (CFL) and gelsolin (GSN) [2], which are involved in Ca^2+^-independent (CFL) and Ca^2+^-dependent (GSN) recycling and remodeling of the actin cytoskeleton. In F-actin, T148 is located in the proximity of the D-loop of the longitudinally adjacent actin subunit, a region forming interaction surfaces with important F-actin-binding proteins. The effects of T148 modification on interaction with these proteins were not yet explored.

Furthermore, pathogenic effects of actin-specific bacterial toxins can be amplified by ABPs, such that major disfunction of the actin cytoskeleton can be caused by modification of only a small fraction of the cellular actin population [16,17]. Yet, previous TccC3 toxicity studies focused only on fully modified actin, using mostly qualitative or semi-quantitative approaches. Since (i) the pathogenetic mechanisms of actin modification by TccC3 toxin can be understood only in the context of ABPs and (ii) the complete modification of actin is unlikely to be achieved in vivo, we expanded the panel of ABPs and evaluated their interaction with partially T148-ADP-ribosylated (ADPR)-actin (Table 1).

First, we found that only the dynamic, tropomyosin (TPM)-free subpopulation of F-actin, not G-actin populations, is effectively modified and, therefore, is likely the physiological substrate of TccC3. Next, we confirmed previous findings that severing and depolymerization of the fully modified actin by CFL were inhibited. The inhibition, however, was minor in F-actin containing a low fraction of modified species. We detected that the affinity of T148-ADPR-G-actin for profilin (PFN1) and TMSB4 were moderately weakened, resulting in mild acceleration of spontaneous polymerization in the presence of these proteins. Interestingly, the Arp2/3 complex-mediated nucleation was also potentiated by the modification. Notably, the modification strongly inhibited actin interaction with tandem calponin-homology (*t*-CH) domain actin organizers resulting in potent inhibition of actin bundling by plastins (PLSs) and α-actinin (ACTN) even with partially modified actin.

Supported by cell imaging experiments, our data suggest that the accumulated disorganized actin clusters upon TccC3-treatment originate from a subpopulation of dynamic F-actin, such as cortical and lamellipodial actin, whose mildly reduced recycling produces G-actin prone to increased spontaneous and Arp2/3-controlled nucleation at undesirable locations. The observed F-actin disorganization is furthered by the resistance of the modified filaments to a higher-order organization by *t*-CH domain proteins, several of which are involved in the organization of actin bundles and networks and the connection of the actin cytoskeleton and the cell membrane.

## 2. Results

### 2.1. Dynamic F-Actin but Not G-Actin Is the Substrate for TccC3 under Physiological Conditions

Actin modification by mART toxins can be monitored by following the increase of fluorescence of nicotinamide 1,N^6^-ethenoadenine dinucleotide (εNAD^+^), a fluorescent analog of nicotinamide adenine dinucleotide (NAD^+^) [18]. Using this approach, we were able to detect efficient modification of F-actin with 5 nM of TccC3, while modification of G-actin required three orders of magnitude higher concentration of the enzyme (i.e., 5 µM) to observe notable yet smaller changes in the fluorescence of εNAD^+^ (Figure 1A and Appendix A).

To confirm that the observed difference accurately reflects the modification rate and to eliminate the possibility that the observed difference reflects the difference in fluorescence intensity of the probe in F-actin compared to in G-actin, we recorded the fluorescence of εADPR-F-actin after its depolymerization by latrunculin B (LatB) (Appendix A). Similarly, we monitored changes in the fluorescence intensity upon polymerization of εADPR-G-actin (Appendix A). In both cases, the original level of fluorescence was not changed, confirming that G-actin is indeed a poor substrate for the TccC3 enzyme compared to F-actin, at least with εNAD^+^ as a co-factor. We also excluded the possibility that Mg^2+^ present in the F-actin buffer might enhance the TccC3 activity: in the presence of 0.5 mM Mg^2+^, the G-actin modification rate remained very low (Appendix A).

Switching to non-fluorescent NAD^+^ as a native substrate of TccC3 and detecting the modification by a difference in electrophoretic mobility by native gel electrophoresis, we discovered that free G-actin can also be modified. However, the addition of either PFN1 or TMSB4 blocked the modification (Figure 1B). Since the complexes of actin with either of the two ABPs represent nearly the entire population of cellular G-actin, we conclude that under physiological conditions, G-actin is not a substrate for TccC3. We also confirmed that NAD^+^ is a preferred substrate over εNAD^+^ for the reaction, as replacing 25% εNAD^+^ with NAD^+^ in the reaction mixture completely abolished the incorporation of the fluorescent nucleotide into F-actin (Appendix A).

For F-actin, the lifetime of actin filaments in different actin assemblies in the cell varies dramatically and to a substantial degree depends on whether the filaments are protected by TPM from the severing activity of actin-depolymerizing factor (ADF)/CFLs. Although TPM is not in direct contact with T148, it may interfere with the binding of TccC3 to the TPM-decorated filaments. Indeed, we found that the modification of actin in the presence of TPM was inhibited 5-fold (Figure 1C), suggesting that only the dynamic F-actin subpopulation (e.g., in the cortex, lamellipodia, and endocytic pits) is the preferred substrate for TccC3, while stress fibers and other actin assemblies protected by TPM are resistant to the modification. Since TPM-decorated F-actin comprises more than 50% (up to 80%) of F-actin in most cells [19], the detected substrate specificity allows channeling the TccC3 toxicity to a minor but most functionally charged subpopulation of actin.

### 2.2. ADP-Ribosylation of T148-Actin Has Only Marginal Effects on the Dynamics of F-Actin

In F-actin, T148 of subunit “i” is located in immediate proximity to the D-loop of longitudinally adjacent subunit “i-2” and particularly its M47 residue [20]. The position of T148 is slightly shifted upon insertion of the D-loop in the hydrophobic cleft during polymerization [20], suggesting that ADP-ribosylation of T148 may interfere with the structure and functional properties of F-actin. We applied the single-particle helical reconstruction algorithm to obtain density maps of fully modified T148-ADPR-F-actin at 3.9 Å resolution (Figure 2A). The obtained density map fits equally well with the recent F-actin reconstructions in either the ADP or AMP-PNP (a non-hydrolyzable ATP analog) states [20,21], except that it contains extra density near T148, accounting for the part of the ADP-ribose moiety (Figure 2B). The distant nucleoside part of the ADP-ribose moiety is poorly represented by electron density, suggesting its flexibility, and leaving uncertainty whether it forms bonds with the “i-2” actin. Yet, the ADP-ribose moiety and D-loop are close, potentially allowing for bonding possibilities (Figure 2B).

Despite this proximity, the thermal stability of neither G- nor F-actin was substantially affected by the modification as assessed by differential scanning fluorimetry (DSF [22]; Figure 3A). Accordingly, spontaneous actin polymerization, monitored via changes in fluorescence intensity of pyrene-actin was only marginally, albeit dose-dependently, inhibited across the entire range of ADPR-actin to total actin ratio, i.e., 0–100% (Figure 3B and Appendix A (actin only)). We detected no measurable influence of the modification on pyrene-actin depolymerization (Figure 3C,D), in agreement with the reported lack of influence on the critical concentration of actin polymerization [2].

### 2.3. ADP-Ribosylation of T148-Actin Inhibits Actin Depolymerization by ADF/CFL

Previous studies pointed on a higher resistance of T148-ADPR-actin to depolymerization by CFL [2] in line with steric clashes imposed by the T148-ADPR group for ADF/CFL binding (Figure 4A). To quantitatively assess the functional effects of ADF/CFLs on F-actin and evaluate their physiological significance, we assessed the binding of ADF to G-actin and monitored the depolymerization of F-actin samples modified to a various extent in the presence of human isoforms of ADF/CFLs. Using the nucleotide exchange rate inhibition by ADF as a reporter, we found that the binding of ADF to the modified G-actin was inhibited ~15-fold (Figure 4B). For depolymerization assays, modified and unmodified G-actin species, each containing 5% pyrene-actin, were co-polymerized at various ratios, and depolymerization of the resultant F-actin (2 µM) was achieved by sequestering the disassembled G-actin by LatB. To approximate the physiological conditions, i.e., ensure that the depolymerization proceeded at the pointed end, a barbed-end capping protein (CP; 100 nM) was added. We found that while ADF/CFL-assisted depolymerization of fully ADPR-actin was potently inhibited, F-actin containing 20% ADPR-actin subunits was depolymerized only slightly slower than unmodified F-actin (Figure 4C–G and Appendix A). Overall, gradual change in the depolymerization efficiency with the modification extent is in agreement with a proportionally inhibited CFL binding and filament severing at the sites containing modified actin along with uninterrupted severing at the remaining sites. Depolymerization induced by CFL1 was the fastest, while that accelerated by ADF was the slowest (Figure 4E–G and Appendix A).

In vertebrate cells, depolymerization from the pointed ends produced by ADF/CFLs is further accelerated by cyclase-associated proteins 1 or 2 (CAP1,2) [23]. If the CAP1,2-assisted detachment of terminal actin subunits is affected by the modification, the inhibition of depolymerization can be more potent. To test this possibility, an N-terminal fragment of CAP1 (N-CAP1) capable of accelerating pointed-end depolymerization [24] was co-incubated with the F-actin mixtures, LatB and CFL1. We found that N-CAP1 moderately increased the rate of depolymerization while preserving the overall gradual dependence on the saturation by the modified actin in the co-polymer (Figure 4H). Overall, while the ADF/CFL-promoted depolymerization of the fully modified F-actin is potently inhibited, depolymerization of partially modified actin is only weakly affected. In the cells, therefore, the depolymerization of partially modified F-actin should produce T148-ADPR-G-actin.

### 2.4. T148-ADP-Ribosylation of G-Actin Impedes the Ability of TMSB4 and PFN1 to Inhibit Spontaneous Nucleation of Actin Filaments

While the physiological substrate of TccC3 is F-actin, T148-ADPR-G-actin can be produced upon depolymerization of the modified F-actin. It has been demonstrated by gel shift on native gel electrophoresis and crosslinking assays that the interactions of T148-ADPR-actin with several G-actin binding proteins (namely, ADF/CFL, TMSB4, fragments of GSN) are inhibited [2]. Since the quantitative analysis of these effects was performed only for TMSB4 [2,11], we revisited this issue by focusing on PFN1 and TMSB4 as major G-actin binding proteins. The affinities of actin binding to PFN1 were evaluated by monitoring changes in the anisotropy of fluorescein-5-maleimide (FITC)-PFN1 upon its binding to actin [25]. The affinities of actin binding to TMSB4 were evaluated by fluorescence anisotropy competition assay against FITC-PFN1. Although actin-PFN1-TMSB4 have been shown to form a ternary complex [26,27], TMSB4 binds to actin ~23 times stronger than to actin-PFN1 complex, allowing us to approximate the affinities without accounting for the formation of the ternary complex. We found that the modification reduces the affinity of TMSB4 to Mg^2+^-G-actin ~6.4-fold, while the affinity of PFN1 was inhibited ~5.2 and 3.8-fold for Mg^2+^- and Ca^2+^-G-actin, respectively (Figure 5A,B).

TMSB4 and PFN1 are critical factors that inhibit spontaneous nucleation in the cell, while TMSB4 also inhibits filament elongation and serves for diffusional delivery of G-actin to the sites of controlled polymerization [28]. Therefore, we examined how T148-ADP-ribosylation affects actin polymerization in the presence of these proteins. For both proteins, the modification reduced their ability to inhibit polymerization rates leading to faster polymerization (Figure 5C,D and Appendix A). The effects were notably stronger for TMSB4, in which case the modification significantly accelerated nucleation and reduced the time at which 50% of actin is polymerized (t_1/2_) from 7343 to 3693 s (for 0 and 100% modified actin, respectively) (Appendix A). A likely explanation for the stronger effects on the polymerization in the presence of TMSB4 despite a similar reduction in its affinity for T148-ADPR-actin is the inhibition of both nucleation and elongation by TMSB4, as compared to inhibition of nucleation only by PFN1. Since the actual cellular concentrations of actin, PFN1, and TMSB4 vary substantially from those used in the in vitro experiments (as well as in different cells and even within the same cell under different stages of cell life [29,30]), the extent of the observed actin polymerization acceleration is also expected to vary.

In both cases, however, the modification promoted spontaneous nucleation of actin filaments, a phenomenon that when out of control can lead to a major disorganization of the actin cytoskeleton due to the formation of filaments at various undesirable locations throughout the cell.

### 2.5. T148-ADP-Ribosylation Promotes Actin Polymerization Nucleated by the Arp2/3 Complex

Spontaneous nucleation of actin is strongly unfavored under physiological conditions. As a result, actin polymerization at the desired locations is coordinated by nucleating proteins, among which the Arp2/3 complex is one of the most abundant and essential in the formation of the dynamic actin pool. The Arp2/3 complex is activated by nucleation promoting factors (NPFs), many of which contain the WH2 actin-binding motif (Wiskott–Aldrich syndrome protein (WASP) homology-2 motif), whose binding to actin mirrors that of the N-terminus of TMSB4 [31] (Figure 6A,B).

We speculated, therefore, that the NPF/Arp2/3-dependent nucleation of T148-ADPR-actin may be inhibited. To our surprise, we found that polymerization rates in the presence of Arp2/3 and N-WASP-VCA (a WH2-containing N-WASP C-terminal domain activating Arp2/3 complex) increased proportionally to the fraction of the modified actin and nearly doubled with fully modified actin, as compared to the unmodified control (Figure 6C,E and Appendix A). In the presence of PFN1, the rate of polymerization was inhibited, but the effects of T148-ADPR-actin on Arp2/3-mediated polymerization were overall comparable (Figure 6D,E and Appendix A). Therefore, in the cell, actin modification by TccC3 should increase not only the rate of actin nucleation in the presence of TMSB4 and PFN1, but also that promoted by the Arp2/3 complex.

### 2.6. Functional Activity of Troponin (TNN) and Select Isoforms of Myosin Are Negatively Affected by T148-ADP-Ribosylation

Myosins are a large family of actin-dependent motor proteins with substantial functional versatility but an overall conserved mode of interaction with the actin filament via their catalytic domains. The binding interface of the myosin catalytic domain on actin includes the D-loop area (Figure 7A) [32]. Accordingly, proteolytic cleavage of the D-loop inhibits the actin-dependent myosin ATPase and actin sliding over surface-attached myosin heads [33]. Kinetic analysis of actin-dependent myosin ATPase activity showed no influence of the T148-ADP-ribosylation in the case of non-muscle myosin-2B (a contractile motor found in stress fibers, cell cortex, and contractile ring), but a moderately increased apparent K_m_ (K_app_) while a slightly decreased V_max_ on a cargo transporter myosin-5B (Figure 7B,C). Since myosin-5B is a processive motor dependent on the integrity of actin subunits on its trajectory, even partial modification may negatively affect the actin-dependent cellular cargo transportation.

In striated muscle, myosin-2 catalytic activity is regulated by Ca^2+^ binding to the TPM/TNN complex. In non-muscle cells, TPMs are critical for actin sorting and stabilization, while the functions of non-muscle TNNs remain controversial. Although TPM in either of its three states is separated from T148 and ADP-ribose, alignment of respective cryo-EM reconstructions (PDB: 6KN7) shows that TNNT and TNNI are immediately close to and may sterically clash with the modified T148 residue (Figure 7D). For TPM/TNN complex, we evaluated the rate of Ca^2+^ dissociation from reconstituted thin filaments following a change in fluorescence of 2-(4′-(iodoacetamido)anilino)naphthalene-6-sulfonic acid (IAANS)-labeled TNNC1 [34] and found it to be accelerated in the presence of fully ADPR-F-actin, as compared to unmodified F-actin (Figure 7E). Although actin ribosylation desensitizes reconstituted thin filaments to Ca^2+^ (as indicated by the faster rate of Ca^2+^ dissociation) similar to other known disease-associated mutations and post-translational modifications [35,36,37], the consequences of this observation on the activation of Ca^2+^-regulated stress fibers are obscure, but likely implies a stronger interaction of the regulatory domain of TNNI with actin [38].

### 2.7. T148-ADP-Ribosylation Impairs F-Actin Crosslinking by t-CH-Domain Actin Cytoskeleton Organizers

Localization of the ADP-ribose moiety along the D-loop of a longitudinally adjacent subunit in F-actin alludes to other possible pathogenic mechanisms via the interference with the binding of ABPs to this area. In particular, D-loop is a part of the interaction surface between actin and members of the large family of the tandem calponin-homology domain (*t*-CH) actin organizers, e.g., spectrin, utrophin, dystrophin, filamin, α-actinin, and plastin [39,40,41,42,43,44,45] (Figure 8A–C). Using differential low-speed centrifugation (17,000× *g*), which allows the separation of F-actin bundles (pellet fraction) from single filaments (supernatant fraction) [46], we found that filaments containing as little as 10% of the T148-ADPR-actin were less effectively bundled by the lymphocyte-specific (PLS2, also known as LCP1; Figure 8D and Appendix A) and ubiquitous (PLS3; not shown) isoforms of human plastin, while almost no bundling occurred when the modified actin content was increased above 20% (Figure 8D,E and Appendix A). Similarly, crosslinking of F-actin by α-actinin (ACTN), another major *t*-CH domain actin-organizing protein, was strongly reduced proportionally to the ADPR-actin fraction in the filament (Figure 8F and Appendix A). Fully modified F-actin was completely resistant to bundling by both proteins even at high concentrations of the crosslinkers (Figure 8E,G bottom panels, Appendix A).

### 2.8. TccC3 Promotes Rapid Disorganization of the Dynamic F-Actin Pool in Living Cells

Unbranched parallel or antiparallel bundles and networks in the cell (e.g., in stress fibers, focal adhesion complexes, contractile rings, filopodia) are stabilized and protected by various isoforms of TPM [47]. On the other hand, branched networks nucleated by the Arp2/3 complex are enriched by proteins (e.g., fimbrins/plastins) competing with TPM [48] and, therefore, are low in TPM. Since TPM strongly inhibit F-actin modification by TccC3 (Figure 1C), it is reasonable to expect that branched networks and their immediate derivatives, as found in lamellipodia, cell cortex, endocytic sites, and other membrane-remodeling regions, should be the most vulnerable to the effects of TccC3, while stress fibers and focal adhesions responsible for cell spreading and adhesion are likely to be more resistant due to TPM protection. To test this hypothesis, we focused on the morphology of HeLa cells affected by various actin cytoskeleton-targeting toxins. Along with TccC3, we tested other toxins affecting actin cytoskeleton: actin-crosslinking domain (ACD) and Rho-inhibitory domain (RID) effectors of *Vibrio cholerae* multifunctional-autoprocessing repeats-in-toxin (MARTX) [49,50,51]. We found that TccC3 caused profound formation of membrane blebs at early stages of toxin treatment (Figure 9; 1 h), reflecting major breaches in the structural integrity of the actin-based cell cortex. At the same time, cells retained their shape and remained firmly attached to the substrate even at longer treatments (i.e., 2 h). In contrast, RID toxin that inhibits actin cytoskeleton-governing small GTPases RhoA, Cdc42, and Rac1 [49] caused profound cell rounding without obvious blebs (Figure 9). Finally, ACD, which creates non-polymerizable covalently crosslinked actin oligomers toxic for several families of tandem/oligomeric G-actin-binding proteins [16,17,50], also caused profound cell rounding and some blebbing, which, however, was notably less prominent than that caused by TccC3 (Figure 9). To summarize, our cellular data supports the prediction that only a subfraction of dynamic F-actin unprotected by TPM, such as found at the cell periphery, is particularly vulnerable to TccC3.

## 3. Discussion

Evolved to overcome the hostile environment of the host immune system, bacterial toxins are typically exceptionally potent and can cause substantial harm at very low doses. High potency represents a particularly major challenge for actin-specific toxins, whose target is the most abundant cellular protein. To compensate for this challenge, the activity of actin-specific toxins is often augmented by actin-binding proteins acting upon only a minor fraction of the modified actin pool [17]. TccC3-hvr is an effector domain of Tc toxin from *P. luminescens* and an actin-specific toxin and, therefore, should obey the above rule, but its toxicity amplification mechanisms are unclear.

In the current study, we revisited the effects of actin ADP-ribosylation at the T148 residue by TccC3. Since T148 is located at the site involved in the interaction with numerous G- and F-actin binding proteins, only a fraction of which was evaluated here, the list of discovered mechanisms is unlikely to be complete. Yet, we believe our findings substantially advance the understanding of the pathogenic mechanisms of the TccC3 toxin, including some of the newly discovered toxicity amplification mechanisms, allowing us to propose a more comprehensive model of the toxicity (Figure 10).

Upon injection via the TcA-formed transmembrane pore to the cytoplasm of host cells, TccC3 attacks the fraction of F-actin that is not protected by TPM isoforms (Figure 1). Since TMSB4- and PFN-bound G-actin also cannot be utilized as substrates for TccC3, this selectivity allows channeling the activity of the enzyme towards a subpopulation of mostly branched lamellipodial F-actin networks and its immediate derivatives—a relatively small actin pool directly involved in essential activities, such as cell migration, endo- and phagocytosis, and membrane and organelle remodeling. When this manuscript was ready for submission, a structural study published as a bioRxiv preprint independently confirmed our finding that F- and not G-actin is a preferable substrate for TccC3 [53]. Using cryo-EM reconstruction as the main approach, the authors found that the TccC3’s binding site on F-actin spans two longitudinally adjacent actin subunits and that the NAD^+^-binding pocket on the enzyme’s active site is shaped by this interaction via an induced-fit mechanism. Since the respective structures are not yet publicly available, our hypothesis that the toxin competes with TPM for actin remains to be validated. We confirmed previous findings that fully modified F-actin is highly resistant to severing and depolymerization by CFL isoforms [2]. However, our data suggests that those effects will not dominate early in the infection, when only a small fraction of actin is modified (Figure 4). At this stage, CFL-mediated recycling of partially modified actin would be affected only weakly, enabling the generation of a monomeric pool of T148-ADPR-actin, whose production would otherwise not be possible either from unmodified G-actin (which is not a preferred TccC3 substrate) or from heavily modified F-actin (which is resistant to ADF/CFL mediated depolymerization). As T148-ADPR-G-actin has a reduced affinity for the two major G-actin-binding partners PFN1 and TMSB4 functioning to prevent spontaneous nucleation, the modification should lead to the initiation of actin polymerization at numerous undesired locations contributing to the disorganization of the cytoskeleton.

Interestingly, not only spontaneous nucleation of T148-ADPR-G-actin but also nucleation by the Arp2/3 complex is potentiated, which should favor the formation of the dynamic F-actin branched network susceptible to the toxin’s activity. The molecular mechanisms behind the improved nucleation by the Arp2/3 complex are puzzling, as the activation of the complex requires WH2 domains of NPFs, e.g., a VCA domain of N-WASP used in this study, but also those of WASH, WHAMM, JMY, and others. WH2 domains of NPFs are homologous to the N-terminal part of TMSB4 and thus their binding to actin is likely to be inhibited by the modification. A possible explanation is that T148-ADP-ribosylation favors recruitment of the Arp2/3 complex to the filament side, which is equally important for priming the complex for nucleation. This speculation is indirectly supported by the observation that out of seven subunits of the Arp2/3 complex, two (ARPC1 and ARPC3) are in a non-clashing proximity to the ADP-ribose moiety on T148 (Figure 6A), and the space between T148-ADP-ribose and both subunits is closing upon the complex activation [54].

T148-ADP-ribosylation of actin has negligible effects on myosin-2, which is found in non-muscle cells mainly in the stress fibers, most of which are well-decorated by TPM and, therefore, should remain protected from the modification. In contrast, the activity of myosin-5 is moderately inhibited by the modification. Myosin-5 is involved in transporting vesicles and cargo through the dense cortical cytoskeleton towards the cytoplasmic membrane. Since the observed inhibition is only partial even in fully modified actin (Figure 7C), it remains to be established whether the modification has a measurable contribution to the pathogenesis. Similarly, a disturbed Ca^2+^ regulation of the troponin complex is relatively minor, and although TNNC is found in non-muscle cells [55], its role in the TccC3 pathogenesis is dubious.

In contrast, we propose that the strong inhibition of actin binding/bundling by *t*-CH-domain proteins (Figure 8) deserves more attention. Indeed, proteins of this family are not only involved in the organization of stress fibers (α-actinins), but intricately implicated in strengthening the actin branched networks in the lamellipodia (e.g., plastins), and connecting the actin cortex with the membrane (e.g., spectrin, utrophin, filamin) and between the actin cytoskeleton components (e.g., filamin, α-actinin). Since the modification does not disrupt or even weaken the filaments per se, the observed massive bleb formation (Figure 9) pointing on the failure of the cortical cytoskeleton to maintain the contact with the cytoplasmic membrane likely results from the weakening of the links between actin and membrane-localized actin-binding proteins, e.g., spectrin. This hypothesis correlates with the observations that disruption of spectrin tetramers in megakaryocytes results in blebbing [56] and that spectrins are cleaved in apoptosis [57,58]. On the other hand, blebbing is more reliably linked to the function of the membrane-to-cytoskeleton tethers from the EMR (ezrin/moesin/radixin) protein family [59], whose interaction with T148-ADPR-actin would be interesting to explore. Alternatively, the enhanced blebbing caused by TccC3 may result not from its more frequent bleb initiation, but from less effective bleb retraction, which depends on the myosin-2-generated force [60] and may require α-actinin to connect antiparallel actin fibers. While the detailed mechanism of enhanced membrane blebbing remains to be determined, the primary targeting of the actin cortex by TccC3 supports the idea that the dynamic population of actin is its primary target.

## 4. Materials and Methods

### 4.1. Protein Purification

#### 4.1.1. TccC3-hvr

cDNA of TccC3-hvr from *P. luminescens* (synthesized by Genscript Biotech, Piscataway, NJ, USA) was cloned with an N-terminal 6xHis-tag into pColdI vector (Takara Bio, San Jose, CA, USA) modified to include a TEV protease recognition site downstream of the 6xHis-tag using NEBuilder HiFi DNA Assembly Master Mix (New England BioLabs, Ipswich, MA, USA). Sequence was verified by Sanger DNA sequencing (Genomics Shared Resource, OSU Comprehensive Cancer Center, Columbus, OH, USA). Recombinant protein was expressed in BL21(DE3)-pLysS *E. coli* (Agilent Technologies, Santa Clara, CA, USA) grown in rich bacterial culture medium (1.25% tryptone, 2.5% yeast extract, 125 mM NaCl, 0.4% glycerol, 50 mM Tris-HCl, pH 8.2). Transformed cells were grown at 37 °C to reach an OD_600_ of 1.2, after which protein expression was induced by adding isopropyl β-D-1-thiogalactopyranoside (IPTG) to 1 mM and switching the growth temperature to 15 °C overnight. 6xHis-TccC3 was bound to HisPur cobalt resin (Thermo Scientific, Rockford, IL, USA) under native conditions and eluted using a buffer containing 250 mM imidazole. Purified TccC3 was dialyzed against a storage buffer containing 10 mM 4-(2-hydroxyethyl)-1-piperazineethanesulfonic acid (HEPES), pH 7.2, 25 mM NaCl, 2 mM dithiothreitol (DTT), 0.1 mM benzylsulfonyl fluoride (PMSF), flash frozen, and stored at −80 °C.

#### 4.1.2. Skeletal Actin

Skeletal actin was purified from rabbit skeletal muscle acetone powder (Pel-Freeze Biologicals, Rogers, AR, USA) or frozen chicken breast muscles (Trader Joe’s) as previously described [61,62]. Actin was stored on ice in G-buffer (5 mM Tris-HCl, pH 8.0, 0.2 mM CaCl_2_, 0.2 mM ATP, 5 mM β-mercaptoethanol (β-ME), 0.1 mM PMSF, 0.005% sodium azide). Pyrenyl-actin was prepared as previously described [17,63] using G-actin devoid of reducing reagent. Unlabeled actin and pyrenyl-actin were further purified by size-exclusion chromatography using HiPrep 26/60 Sephacryl S-200 HR column (Cytiva, Marlborough, MA, USA) for pyrenyl-actin (de)polymerization assays. All actin preparations were stored on ice for up to a month and dialyzed into fresh G-buffer after 2 weeks of storage.

#### 4.1.3. Tropomyosin

Tropomyosin was purified from chicken skeletal muscle (prepared from flash-frozen chicken breast muscles (Trader Joe’s)) as previously described [64]. Eluted tropomyosin was dialyzed against storage buffer (10 mM HEPES, pH 7.0, 30 mM KCl, 2 mM MgCl_2_, 0.5 mM EGTA, 2 mM DTT, 0.1 mM PMSF), flash frozen, and stored at −80 °C.

#### 4.1.4. Cofilins

Recombinant human CFL isoforms and ADF were expressed and purified as previously described [65,66]. Purified proteins were dialyzed against storage buffer (10 mM 3-(N-morpholino) propane-sulfonic acid (MOPS), pH 7.0, 25 mM NaCl, 1 mM DTT, 0.1 mM PMSF), flash frozen, and stored at −80 °C.

#### 4.1.5. N-Terminal Fragment of Cyclase-associated Protein 1

Recombinant human N-CAP1 was expressed and purified as previously described [67]. Eluted N-CAP1 was dialyzed against storage buffer (20 mM Tris-HCl, pH 8.3, 30 mM NaCl, 2 mM DTT, 0.1 mM PMSF), flash frozen, and stored at −80 °C.

#### 4.1.6. Profilin

Recombinant human PFN1 and fluorescein-5-maleimide (FITC)-labeled PFN1(T44C) were prepared as previously described [68]. Cysteine mutation (T44C) was introduced in PFN1 for labeling by Quick-change site-directed mutagenesis following the manufacturer’s protocol (Agilent Technologies, Santa Clara, CA, USA). Following urea elution from poly-L-proline agarose and renaturation, PFN1 was further purified by size-exclusion chromatography using HiPrep 26/60 Sephacryl S-200 HR column (Cytiva, Marlborough, MA, USA). FITC-PFN1(T44C) was prepared by labeling 900 µM DTT-free PFN1(T44C) with 1.2 molar excess of fluorescein isothiocyanate maleimide on ice overnight. To remove free dye, the reaction mixture was passed through a size-exclusion column Superdex 200 Increase 10/300 GL (Cytiva, Marlborough, MA, USA) using buffer containing 20 mM Tris-HCl, pH 8.0, 50 mM NaCl, 1 mM DTT, 0.1 mM PMSF. Purified PFN1 was flash frozen and stored at −80 °C. PFN1 was dialyzed into ATP-free G-buffer before use.

#### 4.1.7. Thymosin-β4

Recombinant rat TMSB4 was expressed and purified as previously described [63]. TMSB4 was dialyzed against pre-G buffer (5 mM Tris-HCl, pH 8.0, 0.2 mM CaCl_2_), flash frozen, and stored at −80 °C. TMSB4 concentration was measured using absorbance at 205 nm with an extinction coefficient of 27 mg^−1^ mL as previously described [69].

#### 4.1.8. Arp2/3 Complex 

Porcine thymus Arp2/3 complex and N-WASP-VCA were prepared as previously described [17].

#### 4.1.9. Plastins

Recombinant human N-terminally 6xHis-tagged PLSs were purified as previously described [44,45]. PLSs were dialyzed against plastin storage buffer (10 mM HEPES, pH 7.0, 30 mM KCl, 2 mM MgCl_2_, 0.5 mM EGTA, 2 mM DTT, 0.1 mM PMSF), flash frozen, and stored at −80 °C.

#### 4.1.10. α-Actinin

ACTN was purified from chicken gizzard as previously described [70]. ACTN was flash frozen and stored at −80 °C in plastin storage buffer.

#### 4.1.11. Capping Protein

Mouse CP heterodimer construct consisting of α1 and β2 CAPZ subunits in pRSFDuet was a gift from Dr. J. Cooper. 6xHis-tagged CP was purified as previously described [71] using TALON metal affinity resin (Takara Bio USA, San Jose, CA, USA), a ceramic hydroxyapatite column (EconoFit CHT type I, 40 μm; Bio-Rad Laboratories Hercules, CA, USA), and a gel-filtration column (Sephacryl S-200 HR; Cytiva/GE Healthcare, Marlborough, MA, USA). The purified CP was stored at −80 °C in a buffer containing 20 mM MOPS, 100 mM KCl, 1 mM Tris(2-carboxyethyl)phosphine (TCEP), pH 7.2.

#### 4.1.12. Myosins

Recombinant human myosin-2B subfragment-1 (S1) and myosin-5B heavy meromyosin (HMM) were overproduced in the baculovirus/Sf9 insect cell system and prepared as previously described [72,73,74]. Proteins were flash frozen and stored at −80 °C in buffer containing 10 mM MOPS pH 7.2, 500 mM NaCl, 0.1 mM EGTA, and 2 mM DTT.

#### 4.1.13. Tropomyosin/troponin complex

Recombinant TPM1, TNNT2, TNNI3, and TNNC1 were bacterially expressed and purified as previously described [37,75]. TNNC1 was labeled with IAANS utilizing the T53C mutation, and the TNN complex and subsequent reconstituted thin filaments were prepared as previously described [34] in 10 mM MOPS, 150 mM KCl, 1 mM DTT, 3 mM Mg^2+^ at pH 7.0.

#### 4.1.14. Protective Antigen (PA)

Recombinant PA (a pore-forming component of the anthrax toxin delivery system [52] used for intracellular delivery of RID, ACD, and TccC3 toxins) was purified as previously described [76]. Eluted PA was further purified using a size-exclusion Superdex 200 Increase 10/300 GL column (Cytiva, Marlborough, MA, USA) in a buffer containing 20 mM Tris-HCl, pH 8.0, 135 mM NaCl, 0.1 mM PMSF. Purified PA was concentrated, flash frozen, and stored at −80 °C.

#### 4.1.15. Fusion Constructs of N-Terminal Fragment of Anthrax Lethal Factor (LF_N_) with Toxins

For intracellular delivery of bacterial toxins, they were fused in-frame with N-terminal LF_N_ (a benign PA-binding component of the anthrax toxin delivery system [52]). TccC3-hvr was cloned with an N-terminal 6xHis-tag followed by LF_N_ into pColdI vector (Takara Bio, San Jose, CA, USA) modified to include a TEV protease recognition site downstream of the 6xHis-tag using NEBuilder HiFi DNA Assembly Master Mix (New England BioLabs, Ipswich, MA, USA). Sequence was verified by Sanger DNA sequencing (Genomics Shared Resource, OSU Comprehensive Cancer Center, Columbus, OH, USA). Recombinant LF_N_-TccC3, LF_N_-ACD, and LF_N_-RID were expressed and purified as previously described [77]. Purified proteins were dialyzed against buffer containing 10 mM HEPES, pH 7.4, 50 mM NaCl, 2 mM DTT, 0.1 mM PMSF, flash frozen, and stored at −80 °C.

### 4.2. ADP-Ribosylation Assays

#### 4.2.1. Fluorescence-Based Assays

Fluorescence-based TccC3-ADP-ribosylation assays were carried out in either G- or F- (10 mM HEPES, pH 7.4, 30 mM KCl, 2 mM MgCl_2_, 0.5 mM EGTA, 2 mM DTT, 0.1 mM PMSF) buffer by adding the following reagents sequentially: 10 µM BSA to stabilize TccC3, 40 µM total NAD^+^ (either 100% of εNAD^+^ (Sigma, St. Louis, MO, USA) or variable ratios of εNAD^+^ to NAD^+^), actin-binding reagents (final concentrations are indicated in the Figure 1C and Appendix A), and 5 µM of either G- or F-actin at 20 °C in a 96-well plate. Fluorescence intensity of εNAD^+^ was recorded at λ_ex_ = 305 nm and λ_em_ = 405 nm using an Infinite M1000 plate reader (Tecan, Männedorf, Switzerland). Following the stabilization of the fluorescence intensity signal, modification reactions were initiated by the addition of 0.08 volumes of TccC3 (to final concentrations of either 5 or 250 nM). The activity of TccC3 was assessed by calculating the tangent slopes of the initial linear range of each εNAD^+^ fluorescence trace.

#### 4.2.2. Electrophoretic Mobility Shift Assays by Native-PAGE

Native-PAGE mobility assays were carried out in the same way as the fluorescence-based assays using non-fluorescent NAD^+^, 5 nM TccC3, and actin-binding reagents (final concentration are indicated in Figure 1B and Appendix A) at 20 °C in Eppendorf tubes. The reactions were stopped by the addition of quenching buffer (10 mM Tris-HCl, pH 8.0, 100 mM L-lactate, 300 mM hydrazine, 60 U/mL L-lactate dehydrogenase (LDH from rabbit muscle; Roche, Mannheim, Germany)) [78]. Quenched reactions were resolved on 10% native polyacrylamide gels devoid of SDS and supplemented with 0.2 mM Ca^2+^ and 0.2 mM ATP. Gel electrophoresis was performed at 100 V using running buffer containing 25 mM Tris-HCl, 192 mM glycine, 0.1 mM Ca^2+^, 0.1 mM ATP, at 4 °C. Densitometry analysis of the bands corresponding to unmodified actin and ADPR-actin was performed using ImageJ software [79] to assess the degree of ADP-ribosylation and the activity of TccC3.

#### 4.2.3. Preparation of 100% TccC3-ADP-Ribosylated Actin

A 100% modification of unlabeled or pyrene-labeled actin (100% ADPR-actin) was achieved by mixing 20–100 µM G-actin with 1.5 molar excess of NAD^+^ and 0.01 molar amount of TccC3 in G-buffer and incubation at 25 °C. After 30 min of incubation, a second dose of equimolar amount of NAD^+^ and 0.01 molar amount of fresh TccC3 was added, and the reactions were supplemented with a third dose of 0.01 molar amount of fresh TccC3 after another 30-min period. The reaction mixtures were then incubated for 1 h at 25 °C and transferred on ice for an overnight incubation. The completeness of ADP-ribosylation was confirmed by native-PAGE, and the reactions were terminated by the removal of the NAD^+^ through three rounds of dialysis against G-buffer. Actin used as “unmodified actin control” (i.e., 0% ADPR-actin) in Figure 3, Figure 4, Figure 5, Figure 6, Figure 7 and Figure 8 was treated under identical conditions in the absence of TccC3. The 100% and 0% T148-ADPR-actin samples were stored on ice for up to a month and dialyzed into fresh G-buffer after 2 weeks of storage. Partially modified actin was prepared by mixing various ratios of 100% and 0% T148-ADPR-actin samples.

For preparation of fully modified TccC3-ADP-ribosylated actin used in the experiments described in Figure 5A,B, the completeness of modification was confirmed by a native gel electrophoresis, and His-tagged TccC3 was removed by passing via HisPur cobalt resin (Thermo Scientific, Rockford, IL, USA) in G-buffer void of reducing agent. The modified actin collected in the flow-through fraction was dialyzed against G-buffer for three rounds to remove NAD^+^. The modified actin was stored in G-buffer on ice for 2 weeks.

### 4.3. Cryo-EM

A sample of TccC3-ADP-ribosylated F-actin was applied to discharged lacey carbon grids and frozen with a Leica EM GP plunge freezer. Movies were collected in a Titan Krios and processed as previously described [44,45]. The overall resolution of the final reconstruction was determined by the Fourier shell correlation (FSC) between two independent half maps. The model of AMP-PNP F-actin (PDB: 6DJM) was fitted into the 3.9 Å resolution cryo-EM map of T148-ADPR-F-actin, followed by examination and adjustment in COOT v.0.8.9.2 [80] The manually curated model was then real-space refined in PHENIX v.1.19 [81]. The cryo-EM map was deposited with accession code EMD-26987 in the Electron Microscopy Data Bank (EMDB).

### 4.4. Differential Scanning Fluorimetry (DSF)

Actin was diluted to 5 µM in either G- or F-buffer (G-buffer supplemented with 30 mM KCl, 2 mM MgCl_2,_ and 0.5 mM EGTA) in the presence of 1x Sypro Orange dye (Invitrogen, Waltham, MA, USA). Changes in fluorescence of the dye with increasing temperature were recorded using CFX real-time PCR detection system (Bio-Rad Laboratories, Hercules, CA, USA). The first derivatives of fluorescence intensity against temperature (dF/dT) were calculated and normalized; the melting temperatures (T_m_) were determined as the maximum of the first derivative.

### 4.5. Pyrenyl-Actin-Based Polymerization and Depolymerization Assays

Pyrenyl-actin (de)polymerization assays were performed as described [16]. To prepare unmodified or 100% T148-ADPR-G-actin samples containing 5, 10, or 30% pyrene-actin, unlabeled 0 or 100% modified actins were mixed with pyrene-labeled 0 or 100% modified actins. The G-actin mixtures were used in the polymerization assays or pre-polymerized for 30 min by the addition of 30 mM KCl, 2 mM MgCl_2_, and 0.5 mM EGTA for the depolymerization assays. Pyrene fluorescence intensity was recorded at λ_ex_ = 365 nm and λ_em_ = 407 nm using either an Infinite M1000 plate reader (Tecan, Männedorf, Switzerland) for the polymerization assays and LatB-induced depolymerization assays, or a QuantaMaster spectrofluorometer (Horiba, Kyoto, Japan) for the dilution-induced depolymerization assays. The initial nucleation rate, the time at which 50% of the actin was polymerized (t_1/2_), and the actin polymerization rate at t_1/2_ were calculated as previously described [82].

### 4.6. Fluorescence-Based Binding Assays

For the fluorescence anisotropy binding assays, 100 nM FITC-PFN1(T44C) in G-buffer (Ca^2+^ condition) or G-buffer with 0.5 mM EGTA and 0.12 mM MgCl_2_ (Mg^2+^ condition) was mixed with G-actin (0.025–20 μM) and incubated for 45 min at 25 °C (Mg^2+^ condition) or incubated overnight at 4 °C followed by an incubation for 9 h at 25 °C (Ca^2+^ condition). Fluorescence anisotropy was recorded at λ_ex_ = 470 nm and λ_em_ = 518 nm using an Infinite M1000 plate reader (Tecan, Männedorf, Switzerland).

The relative fluorescence anisotropy changes were normalized and fitted to the binding isotherm equation using Origin software (OriginLab, Northampton, MA) to estimate the binding affinity of actin to the FITC-PFN1(T44C):(1)$$\frac{I}{I_{\max}}=\frac{P+X+K_{d}-\sqrt{{\left(P+X+K_{d}\right)}^{2}-4\mathrm{PX}}}{2P},$$
where I is the relative fluorescence anisotropy, I_max_ is the maximum change in fluorescence anisotropy, P is the concentration of FITC-PFN1(T44C), X is the concentration G-actin, and K_d_ is the dissociation constant.

For the fluorescence anisotropy competition assays, 6 µM FITC-PFN1(T44C) was pre-mixed with 6 µM actin or 12 µM 100% ADPR-actin in G-buffer and incubated for 45 min at 25 °C. PFN1-actin mixture was diluted 6-fold in the solution containing TMSB4 (0.25–150 µM final concentration) in G-buffer supplemented with 0.5 mM EGTA and 0.12 mM MgCl_2_) and further incubated for 1 h at 25 °C. Fluorescence anisotropy was recorded at λ_ex_ = 470 nm and λ_em_ = 518 nm using an Infinite M1000 plate reader (Tecan, Männedorf, Switzerland).

The relative fluorescence anisotropy changes were normalized to the calculated bound fractions (0.60 for actin and 0.52 for modified actin) and fitted to the following equation [83] using Origin software (OriginLab, Northampton, MA, USA) to estimate the binding affinity of actin to the TMSB4:

Fraction PFN1 bound
(2)$$=\frac{-K_{d2}K_{d1}-2K_{d2}A-K_{d1}X+K_{d1}A+\sqrt{K_{d2}^{2}K_{d1}^{2}+2K_{d2}K_{d1}^{2}X+2K_{d2}K_{d1}^{2}A+K_{d1}^{2}X^{2}-2K_{d1}^{2}\mathrm{AX}+K_{d1}^{2}A^{2}}}{2\left(-K_{d1}K_{d2}-K_{d2}A-K_{d1}X+K_{d1}^{2}+K_{d1}A\right)},$$
where A is the total concentration of G-actin, X is the total concentration of TMSB4, K_d1_ is the dissociation constant for FITC-PFN1(T44C) binding to G-actin, and K_d2_ is the dissociation constant for TMSB4 binding to G-actin.

### 4.7. Nucleotide Exchange Assays

Nucleotide exchange assays were performed as previously described [84] in reaction buffer containing 5 mM HEPES, pH 7.5, 0.1 mM MgCl_2_, 0.4 mM EGTA, 1 mM DTT, 0.2 mM PMSF. Then, 0.2 μM εATP-actin was mixed with 0.25–7.5 μM ADF. The reaction was initiated by adding ATP to 100 μM final concentration. The decrease of εATP fluorescence upon its release from actin was recorded at λ_ex_ = 350 nm and λ_em_ = 412 nm on an Infinite M1000 plate reader (Tecan, Männedorf, Switzerland) and fitted to a single exponential decay equation using Origin software (OriginLab, Northampton, MA, USA). The normalized decay rates plotted against ADF concentration were fitted to the binding isotherm Equation (1) to estimate the binding affinity of actin to the actin-binding proteins.

### 4.8. F-Actin Bundling Assays

Low-speed actin co-sedimentation assays were employed to test F-actin bundling by actin crosslinkers as previously described [45]. Partially modified G-actin samples were prepared by mixing various ratios of 0 and 100% T148-ADPR-G-actin before polymerization. G-actin was treated with 0.1 mM MgCl_2_ and 0.5 mM EGTA on ice for 10 min to switch from Ca^2+^- to Mg^2+^-bound actin state. Subsequently, HEPES (pH 7.0), MgCl_2,_ and KCl were added to final concentrations of 10, 2, and 30 mM to initiate actin polymerization, which was carried out at 25 °C for 30 min. For F-actin bundling by PLSs, 5 µM F-actin in PLS storage buffer (see PLS purification section) was titrated with 0.5–10 µM PLS. For F-actin bundling by ACTN, 2 µM F-actin in PLS storage buffer was titrated with 0.01–4 µM ACTN. Reactions were incubated at 4 °C overnight followed by 4 h at 25 °C. The samples were subjected to centrifugation at 17,000× *g*, at 25 °C for 15 min. Supernatants containing single actin filaments were collected and supplemented with 4× reducing sample buffer. Pellets containing F-actin bundles were soaked in the original volume of PLS buffer supplemented with 4x reducing sample buffer and resuspended by vigorous pipetting. The fractions were subjected to SDS-PAGE for densitometry analysis.

Densitometry analysis of the Coomassie-stained gel bands corresponding to actin was performed using ImageJ software to determine the fraction of bundled F-actin. Bundling efficiency of PLS2 and ACTN was calculated by fitting the data to the Hill equation using Origin software (OriginLab, Northampton, MA, USA):(3)$$\mathrm{Fraction}\;\mathrm{of}\;\mathrm{actin}\;\mathrm{bundled}=\frac{{\left[\mathrm{Actin}\;\mathrm{crosslinker}\right]}^{n}}{{\mathrm{BC}}_{50}{}^{n}+{\left[\mathrm{Actin}\;\mathrm{crosslinker}\right]}^{n}},$$
where n is the Hill coefficient and BC_50_ is the concentration of the actin crosslinker at 50% F-actin bundled.

### 4.9. Myosin ATPase Activity Assays

The actin-activated steady-state ATPase activity was assessed by NADH- (reduced nicotinamide adenine dinucleotide) coupled myosin ATPase assays in 10 mM MOPS (pH 7.0), 2 mM ATP, 50 mM NaCl, 2 mM MgCl_2_, 0.15 mM EGTA, 40 units/mL L-lactic dehydrogenase, 200 units/mL pyruvate kinase, 200 μM NADH, and 1 mM phosphoenolpyruvate at myosin concentrations of 20 nM (Myosin-5B-HMM) or 400 nM (Myosin-2B-S1) at a temperature of 25 °C as previously described [73,74]. Changes in absorbance at 340 nm (ϵ = 6220 M^−1^ cm^−1^) were observed in a Spectramax iD5 multi-mode microplate reader (Molecular Devices, San Jose, CA, USA).

### 4.10. Thin Filament Functional Assays

The working buffer used for the kinetic measurements was 10 mM MOPS, 150 mM KCl, 1 mM DTT, 3 mM MgCl_2_, at pH 7.0. Then, 10 mM EGTA was utilized to remove 200 µM Ca^2+^ from the thin filaments. Ca^2+^ dissociation rates were measured using a stopped-flow instrument (model SX.20; Applied Photophysics, Leatherhead, UK) with a dead time of 1.4 ms at 15 °C [85]. IAANS fluorescence was excited at 330 nm and monitored through a 510 nm broad band-pass interference filter (Oriel, Stratford, CT, USA). Data traces (an average of 3 to 5 individual traces) were fitted with a single exponential equation to calculate the kinetic rates.

### 4.11. Cell Culture

HeLa cells were cultured in Dulbecco’s Modified Eagle Medium (Corning, Corning, NY, USA) supplemented with 1% penicillin–streptomycin (Cytiva/HyClone, Marlborough, MA, USA) and 10% fetal bovine serum (Corning, Corning, NY, USA) at 37 °C with 5% CO_2_ in a humidified incubator. The identity of the HeLa cells was verified by Short Tandem Repeat (STR) profiling (Amelogenin + 9 loci) at the Genomic Shared Resource (OSU, Comprehensive Cancer Center, Columbus, OH, USA). The cells were mycoplasma-negative as determined by PCR tests [86]. Cells were treated with a mixture of PA with one of the LF_N_-fused toxins (2.5 molar excess of PA to an LF_N_-toxin: LF_N_-RID, LF_N_-ACD, LF_N_-TccC3) and imaged using phase contrast on Nikon Eclipse Ti-E microscope equipped with Nikon CFI Plan Apo λ × 20 objective (NA 0.75) and Nikon DS-Qi1Mc camera (Nikon Instruments, Melville, NY, USA).

## Figures and Tables

**Figure 1 ijms-23-07026-f001:**
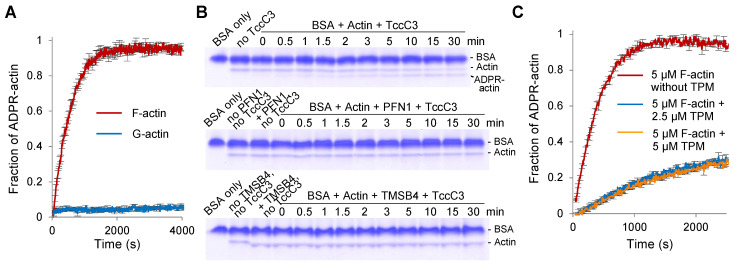
The dynamic pool of F-actin is the physiological substrate of TccC3. (**A**,**C**) G- and/or F-actin ADP-ribosylation by TccC3 in the presence of bovine serum albumin (BSA; added for TccC3 stabilization) was monitored by an increase in εNAD^+^ fluorescence. The reactions were initiated by 5 nM of TccC3. Normalized intensity plots are shown; error bars are shown for every third data point and represent standard errors of the mean of three independent repetitions. In (**C**), the reactions were conducted in the absence or presence of different concentrations of tropomyosin (TPM). (**B**) Representative native-PAGE gels from three independent ADP-ribosylation assays using either G-actin alone (upper panel) or in the presence of either profilin (PFN1; middle panel) or thymosin β4 (TMSB4; lower panel). Uncropped gels are shown in Appendix A.

**Figure 2 ijms-23-07026-f002:**
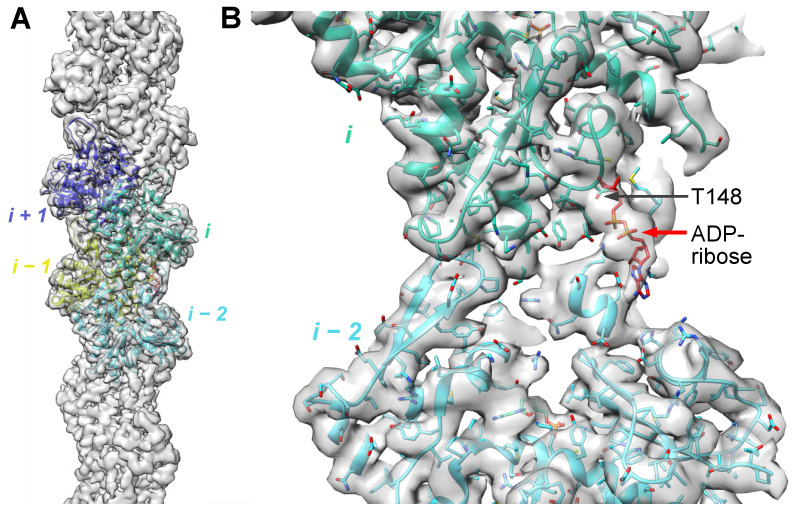
Cryo-EM map of T148-ADPR-F-actin. (**A**) The atomic model of AMP-PNP F-actin (PDB: 6DJM) was fitted into the 3.9 Å resolution cryo-EM map of T148-ADPR-F-actin (present study; EMD-26987). Actin subunits (i + 1, i, i − 1, i − 2) (**A**,**B**) and the T148-ADPR (**B**) are indicated in the subunit “i”.

**Figure 3 ijms-23-07026-f003:**
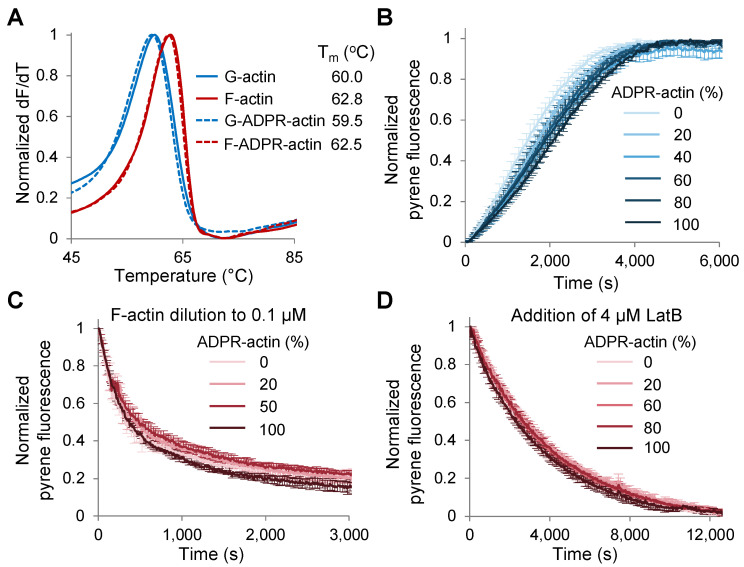
T148-ADP-ribosylation has marginal effects on F-actin thermal stability and dynamics. (**A**) DSF melting profiles and melting temperatures (T_m_s) of unmodified and TccC3-ADP-ribosylated-G- and F-actin; the curves represent the averages of four independent repetitions. (**B**) Spontaneous polymerization of pyrene-labeled (5%) G-actin with various content of T148-ADPR-actin was monitored by an increase in pyrene fluorescence. Normalized intensity plots are shown; error bars represent standard errors of the mean of three independent repetitions and are shown for every third data point for clarity. (**C**,**D**) Depolymerization of preassembled actin filaments. Normalized intensity plots are shown; error bars are shown for every third data point and represent standard errors of the mean of three (**C**) and two (**D**) independent repetitions. (**C**) Depolymerization of pyrene-labeled (30%) F-actin was initiated by diluting F-actin to 0.1 µM. (**D**) Depolymerization of pyrene-labeled (10%) F-actin in the presence of capping protein (50 nM) was initiated by adding latrunculin B (LatB; 4 µM).

**Figure 4 ijms-23-07026-f004:**
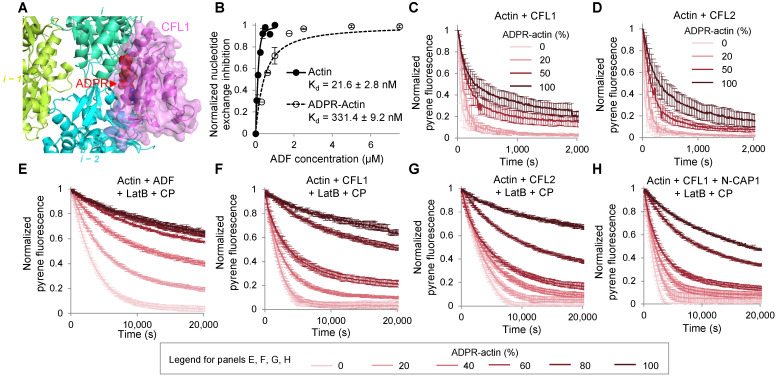
TccC3-ADP-ribosylation inhibits actin depolymerization by ADF/CFL. (**A**) Superimposed structures of T148-ADPR-F-actin (present study; EMD-26987) and CFL1-F-actin (PDB ID: 6VAO). (**B**) The binding of ADF to unmodified and TccC3-ADP-ribosylated-actin was assessed by nucleotide exchange assays. Normalized data are shown; error bars represent standard errors of the mean of two independent repetitions. Two-tailed Student’s *t*-test was used for data comparison; *p*-value is 0.011. (**C**–**H**) Depolymerization of preassembled actin filaments. Normalized intensity plots are shown; error bars are shown for every fifth data point and represent standard errors of the mean of two or three independent repetitions. (**C**,**D**) Depolymerization of pyrene-labeled (30%) F-actin in the presence of 0.2 µM of either CFL1 (**C**) or CFL2 (**D**) was initiated by diluting F-actin to 0.1 µM. (**E**–**H**) Depolymerization of preassembled pyrene-labeled (5%) F-actin in the presence of 250 nM ADF (**E**), 250 nM CFL1 (**F**), 250 nM CFL2 (**G**), or 250 nM CFL1 with 750 nM N-CAP1 (**H**) was initiated by adding latrunculin B (4 µM) and capping protein (100 nM).

**Figure 5 ijms-23-07026-f005:**
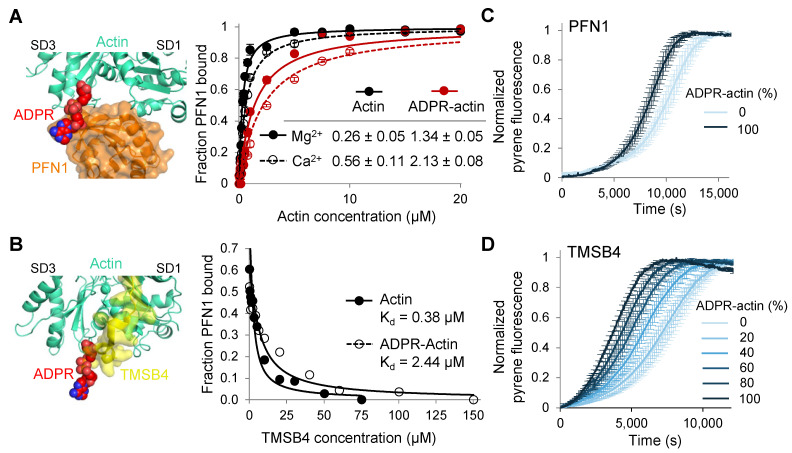
T148-ADP-ribosylation impedes the ability of profilin (PFN1) and thymosin β4 (TMSB4) to inhibit spontaneous actin nucleation. (**A**) Binding of PFN1 to unmodified and TccC3-ADP-ribosylated actin was assessed by changes in fluorescence anisotropy using FITC-labeled PFN1(T44C). Normalized data are shown; error bars represent standard errors of the mean of three independent repetitions. Two-tailed Student’s *t*-test was used for data comparison; *p*-value is 0.00001 and 0.00006 for Mg^2+^ and Ca^2+^ conditions, respectively. T148-ADPR-actin structure (present study; EMD-26987) superimposed with PFN1-actin structure (PDB: 2BTF) is shown on the left. (**B**) Binding of thymosin β4 (TMSB4) to unmodified and T148-ADPR-actin was assessed by fluorescence anisotropy competition assay with FITC-labeled PFN1(T44C). Normalized data from a single experiment are shown. T148-ADPR-actin structure (present study; EMD-26987) superimposed with TMSB4-actin structure (PDB: 4PL7) is shown on the left. (**C**,**D**) Polymerization of 2.5 µM pyrene-labeled (5%) G-actin with various content of T148-ADPR-actin in the presence of 3 µM PFN1 (**C**) or 2.5 µM TMSB4 (**D**) was monitored by an increase in pyrene fluorescence. Normalized intensity plots are shown; error bars are shown for every fifth data point and represent standard errors of the mean of three independent repetitions.

**Figure 6 ijms-23-07026-f006:**
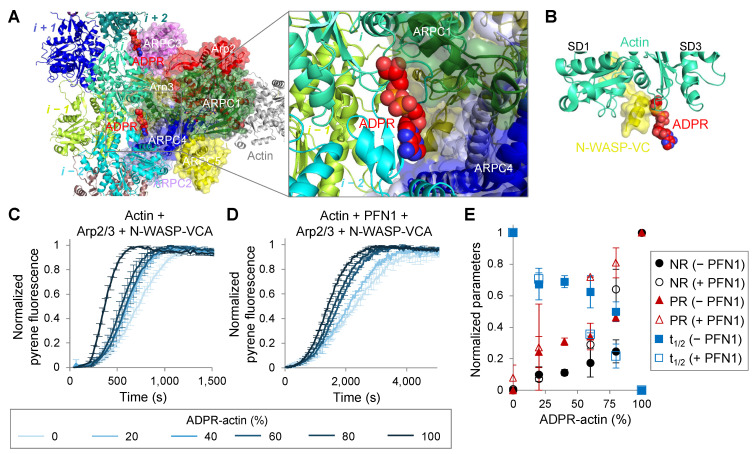
T148-ADP-ribosylation promotes Arp2/3-nucleated actin polymerization. (**A**) Superimposed structures of T148-ADPR-F-actin (present study; EMD-26987) and Arp2/3-F-actin (PDB ID: 7TPT). A zoomed view is shown on the right. (**B**) Superimposed structures of T148-ADPR-F-actin (present study; EMD-26987) and N-WASP (VC domain)-actin (PDB ID: 2VCP). (**C**,**D**) Polymerization of 2 µM pyrene-labeled (5%) G-actin with various content of T148-ADPR-actin in the presence of 50 nM Arp 2/3 complex and 50 nM N-WASP-VCA without (**C**) or with 4 µM PFN1 (**D**) was monitored by an increase in pyrene fluorescence. Normalized intensity plots are shown; error bars are shown for every third data point and represent standard errors of the mean of three independent repetitions. (**E**) Normalized initial nucleation rate (NR), the time at which 50% of actin is polymerized (t_1/2_), and actin polymerization rate at t_1/2_ (PR) were calculated from the polymerization plots shown in (**C**,**D**) and plotted against the percentage of ADPR-actin. Error bars represent standard errors of the mean of three independent repetitions.

**Figure 7 ijms-23-07026-f007:**
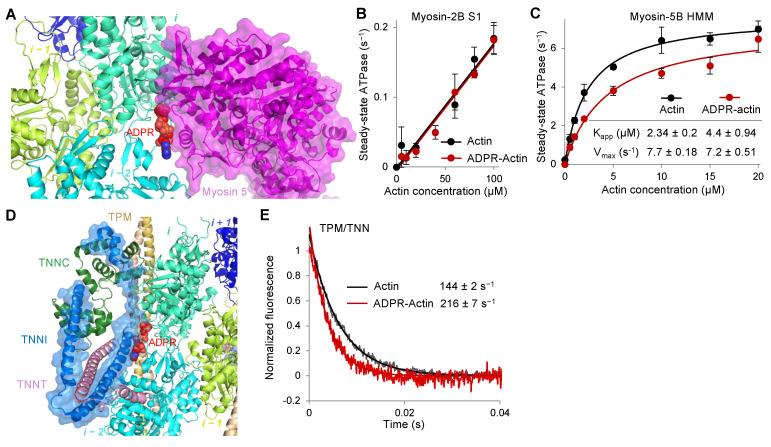
T148-ADP-ribosylation regulates the functions of myosin and TPM. (**A**) Superimposed structures of T148-ADPR-F-actin (present study; EMD-26987) and myosin-5-F-actin (PDB ID: 7PLT). (**B**,**C**) Myosin ATPase activities of myosin-2B subfragment-1 (S1) (**B**) and myosin-5B heavy meromyosin (HMM) (**C**) conducted using unmodified and TccC3-ADP-ribosylated actin. (**D**) T148-ADPR-F-actin structure (present study; EMD-26987) superimposed with TPM/TNN/F-actin structure (PDB: 6KN8). (**E**) Ca^2+^ release from reconstituted thin filaments using unmodified and T148-ADPR-actin assessed by IAANS-labeled TNNC1.

**Figure 8 ijms-23-07026-f008:**
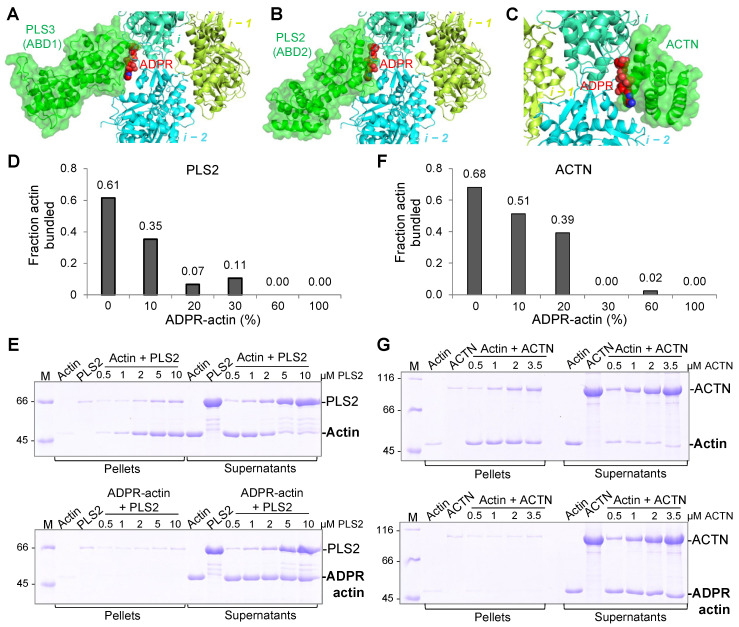
T148-ADPR-actin cannot be bundled by plastin 2 (PLS2) or α-actinin (ACTN). (**A**–**C**) Superimposed structures of T148-ADPR-F-actin (present study; EMD-26987) with either PLS3-ABD1 (PDB ID: 1AOA) fitted to a cryo-EM map (EMD-25371) (**A**), PLS2-ABD2-F-actin (PDB ID: 6VEC) (**B**), or ACTN-F-actin (PDB ID: 3LUE) (**C**). (**D**–**G**) F-actin bundling by PLS2 (**D**,**E**) or ACTN (**F**,**G**) was assessed by low-speed actin co-sedimentation assays. (**D**,**F**) Data quantified from single repetition from SDS-PAGE gels are shown in Appendix A (for **D**) and Appendix A (for **F**) using various percentages of T148-ADPR-F-actin with either 5 µM F-actin and 2 µM PLS2 (**D**) or 2 µM F-actin and 0.65 µM ACTN (**F**). (**E**,**G**) Representative SDS-gels resolving bundled (pellet fractions) and non-bundled (supernatant fractions) F-actin in the presence of PLS2 (**E**) or ACTN (**G**) using either unmodified (top panels) or 100% TccC3-ADP-ribosylated F-actin (bottom panels). Uncropped gels are shown in Appendix A.

**Figure 9 ijms-23-07026-f009:**
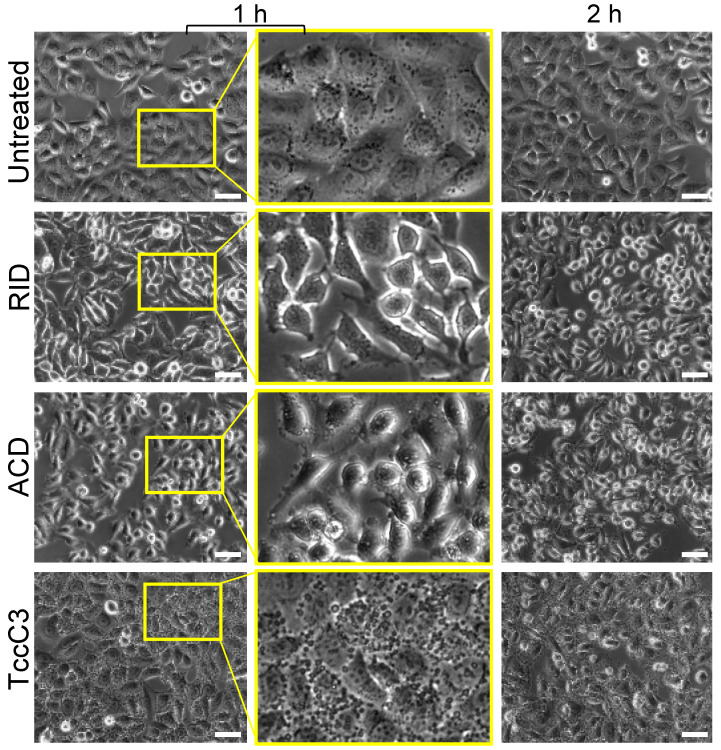
TccC3 induces excessive cell blebbing. Representative micrographs of HeLa cells treated with various bacterial toxins: RID, ACD, and TccC3 at final concentrations of 1, 1, and 0.5 nM, respectively. The toxins were delivered to the cells using anthrax toxin delivery system utilizing protective antigen and the N-terminal domain of lethal factor genetically fused to the toxin of interest [52]. Boxed areas at 1 h of treatment are enlarged (middle panels). Scale bars are 50 µm.

**Figure 10 ijms-23-07026-f010:**
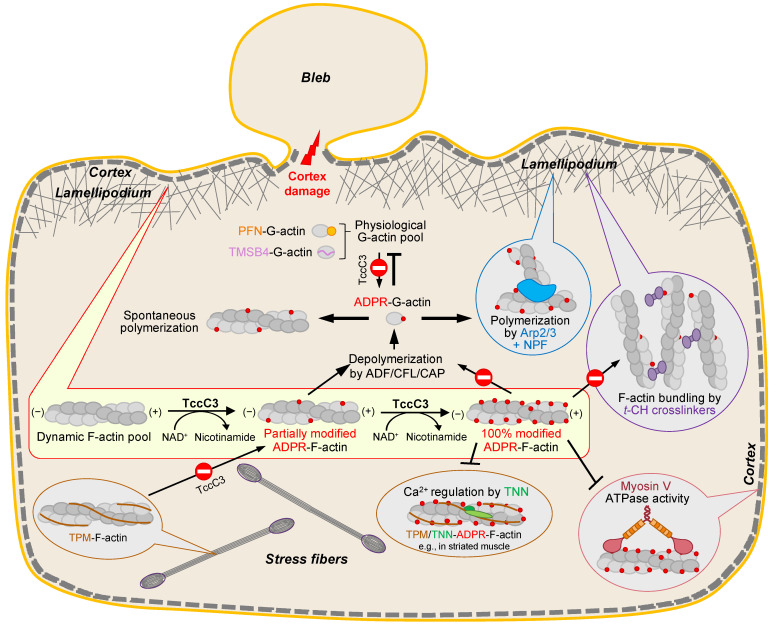
A proposed model of TccC3 toxicity (see explanation in the text).

**Table 1 ijms-23-07026-t001:** Proteins analyzed or discussed in the article. Full and abbreviated names of the proteins are provided in alphabetical order.

Abbreviations	Names
ABD	Actin-binding domain
ACD	Actin-crosslinking domain of MARTX toxin
ACTN	α-actinin
ADF	Actin-depolymerizing factor
ADPR-actin	T148-ADP-ribosylated-actin
BSA	Bovine serum albumin
CAP	Cyclase-associated protein
CFL	Cofilin
CP	Capping protein
EMR	Ezrin/moesin/radixin
F-actin	Filamentous actin
G-actin	Globular actin
GSN	Gelsolin
LCP1	Lymphocyte cytosolic protein 1 (also known as PLS2)
mART	Mono ADP-ribosyltransferase
MARTX	Multifunctional-autoprocessing repeats-in-toxin
NPF	Nucleation-promoting factor
PFN1	Profilin 1
PLS2	Plastin 2
PLS3	Plastin 3
RID	Rho-inhibitory domain of MARTX toxin
Tc	Toxin complex
*t*-CH domain	Tandem calponin-homology domain
TMSB4	Thymosin-β4
TNN	Troponin
TPM	Tropomyosin
WASP	Wiskott–Aldrich syndrome protein
WH2	WASP homology 2 domain

## Data Availability

The cryo-EM map of T148-ADPR-F-actin was deposited with accession code EMD-26987 in the EMDB.

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
