# Peer review of "Photorhabdus luminescens* TccC3 Toxin Targets the Dynamic Population of F-Actin and Impairs Cell Cortex Integrity"

_ijms, 2022, doi:10.3390/ijms23137026_

Round 1

Reviewer 1 Report

In this paper, Dong et al. describe how the TccC3 toxin by ADP-ribosylating actin at Threonine 148 regulates actin dynamics. First, they found that the toxin can modify the filaments only if they are unprotected. Then, they examined, by combining biophysical methods and structural analysis, how this modification controls the interaction with different Actin Binding Proteins (ABPs) and thus the impact on the associated functions in actin assembly. Afterwards, they observed that TccC3 induced high cell blebbing. Finally, based on all the results, they presented a model of TccC3 toxicity.

The roles of TccC3 toxin on actin dynamics have been well studied in the past. As the authors state in the manuscript Thr148-ADP-ribosylated actin favors polymerization, and is resistant to depolymerization by ADF and cofilin. It has reduced affinity for profilin and thymosin-ß4. These functions are confirmed and well-characterised in this study by varying the fraction of modified actin. Various innovative aspects are highlighted such as the promotion of actin polymerization nucleated by the Arp2/3 complex, the regulation of myosin 5B and the antibuddling activity.

Some points need to be clarified by adding more models and experiments. I could support publications after the following revisions have been addressed.

Major points:

As described by De La Cruz et al, Biophysical Journal, 2000 (reference 25), the Kd of the actin- thymosin β4 complex depends on the cation and nucleotide bound to actin. You have performed your experiments in G-buffer (Ca2+) in Figure 5, How does the affinity vary in the presence of Mg2+? Same question for profilin.

Does Threonine 148-ADP-ribosylation continue to impede the ability of thymosin β4 and profilin to inhibit actin nucleation when they are at higher concentrations ? Can you add a dependent dose?

To clarify the effect of Threonine 148-ADP-ribosylation on the actin polymerisation nucleated by the Arp2/3 complex in Figure 6, a single filament observation experiment using TIRF microscopy seems necessary. This will allow to distinguish between the effects on nucleation and branching. In addition, a structural model with a Arp2/3 branched filament with the indication of the Threonine 148-ADP-ribosylated should be added in figure 6 to prove that the hypothesis proposed in the discussion is possible.

Structural models should be added in Figure 8 to show the binding sites of plastin 2 and alpha actinin on F actin.

Minor points:

The concentration of profilin used in Figure 6B must be added.

An additional figure to represent the normalised data in Figure 6C will provide a better understanding. The different information should be separated as in Figure S3.

The gels corresponding to the experiments with the different fractions of modified actin (Figure 8A and C) should be added in a supplementary figure.

Author Response

Please find our point-by-point response to the reviewer’s comments below in green.

Reviewer #1:

Comments and Suggestions for Authors

In this paper, Dong et al. describe how the TccC3 toxin by ADP-ribosylating actin at Threonine 148 regulates actin dynamics. First, they found that the toxin can modify the filaments only if they are unprotected. Then, they examined, by combining biophysical methods and structural analysis, how this modification controls the interaction with different Actin Binding Proteins (ABPs) and thus the impact on the associated functions in actin assembly. Afterwards, they observed that TccC3 induced high cell blebbing. Finally, based on all the results, they presented a model of TccC3 toxicity.

The roles of TccC3 toxin on actin dynamics have been well studied in the past. As the authors state in the manuscript Thr148-ADP-ribosylated actin favors polymerization, and is resistant to depolymerization by ADF and cofilin. It has reduced affinity for profilin and thymosin-ß4. These functions are confirmed and well-characterised in this study by varying the fraction of modified actin. Various innovative aspects are highlighted such as the promotion of actin polymerization nucleated by the Arp2/3 complex, the regulation of myosin 5B and the antibuddling activity.

Some points need to be clarified by adding more models and experiments. I could support publications after the following revisions have been addressed.

We would like to thank the reviewer for the careful reading and positive evaluation of our work. We conducted additional experiments as specified below and addressed all other suggestions of the reviewer. We hope that the manuscript will be now acceptable for publication.

Major points:

As described by De La Cruz et al, Biophysical Journal, 2000 (reference 25), the Kd of the actin- thymosin β4 complex depends on the cation and nucleotide bound to actin. You have performed your experiments in G-buffer (Ca2+) in Figure 5, How does the affinity vary in the presence of Mg2+? Same question for profilin.

We agree that Mg2+-actin is more physiologically relevant substrate for Thymosin β4 and profilin. Therefore, we repeated the experiments with Mg2+ actin, which now replace the previous Ca2+-actin experiments in Fig. 5A,B. For profilin, the difference in affinities is slightly higher for Mg2+- than Ca2+-actin (5.2 fold and 3.8 fold, respectively) when the modified and unmodified actins are compared. Similarly, for TMSB4, the binding affinity under Mg2+ condition measured via competition with profilin was found to be ~6.4 fold, which is comparable to the 7-fold difference reported previously [Lang AE, Schmidt G, Schlosser A, Hey TD, Larrinua IM, Sheets JJ, Mannherz HG, Aktories K. Photorhabdus luminescens toxins ADP-ribosylate actin and RhoA to force actin clustering. Science. 2010 Feb 26;327(5969):1139-42. doi: 10.1126/science.1184557. PMID: 20185726].

Does Threonine 148-ADP-ribosylation continue to impede the ability of thymosin β4 and profilin to inhibit actin nucleation when they are at higher concentrations? Can you add a dependent dose?

In the manuscript, the actin polymerization assay was conducted in the presence of 2.5 µM actin and 3 µM PFN1 or 2.5 µM TMSB4. Given the ADP-ribosylation of T148 weakens but does not abolish binding of these proteins, we predict that high enough concentrations of either PFN1 or TMSB4 can compensate for the reduced affinities and the proteins should more effectively inhibit actin polymerization under such conditions. This, however, does not cancel the modification-caused potentiation of nucleation and polymerization rates in principle. The obtained and provided Kd values allow prediction of the proteins’ behavior under various conditions. Reproducing the exact cellular conditions in vitro is impractical and even impossible, as the levels of G- and F-actins, as well as levels of PFN1 and TMSB4 vary not only between cells but also within the same cell at different stages of its life cycle. The following sentence explaining these thoughts has been added to the manuscript (page 9):

“Since the actual cellular concentrations of actin, PFN1, and TMSB4 vary substantially from those used in the in vitro experiments (as well as in different cells and even within the same cell under different stages of cell life), the extent of the observed actin polymerization acceleration is also expected to vary.”

To clarify the effect of Threonine 148-ADP-ribosylation on the actin polymerisation nucleated by the Arp2/3 complex in Figure 6, a single filament observation experiment using TIRF microscopy seems necessary. This will allow to distinguish between the effects on nucleation and branching. In addition, a structural model with a Arp2/3 branched filament with the indication of the Threonine 148-ADP-ribosylated should be added in figure 6 to prove that the hypothesis proposed in the discussion is possible.

Structural models should be added in Figure 8 to show the binding sites of plastin 2 and alpha actinin on F actin.

We thank the review for these suggestions. The requested structural models have been added (new Figures 6A,B, 7A, and 8A-C.

We also attempted to perform the requested TIRFM experiments, which preliminary did not produce explanation to the observed acceleration. It is expected for the following reasons:

1) With the exception of a very special NPF Dip1, all other NPFs (including the VCA domain of WASP used in our assays) promote actin nucleation by branching from existing actin filaments. Therefore, we expected and preliminary confirmed the branching type of nucleation for both modified and unmodified actins.

2) The mild acceleration of polymerization of the modified actin is difficult to reliably document by TIRFM, which, by nature, is a less quantitative/more labor-intense method for the assemble-level effects as compared to pyrene-actin fluorescence. Nevertheless, we tentatively observed more branches at the pointed ends of T148-modified F-actin, potentially accounting for the difference documented on Figure 6. It would require much more work, though, to reliably confirm the difference, which task extends beyond the goals of the current manuscript and will be pursued as a separate study.

Minor points:

The concentration of profilin used in Figure 6B must be added.

The concentration of profilin used in Figure 6D (former Figure 6B corresponding to the new Figure 6D) has been added to the corresponding Figure legend.

An additional figure to represent the normalised data in Figure 6C will provide a better understanding. The different information should be separated as in Figure S3.

New Supplementary Figure S5 has been added to provide the requested information.

The gels corresponding to the experiments with the different fractions of modified actin (Figure 8A and C) should be added in a supplementary figure.

New Supplementary Figures S6 and S7 have been added to include the requested gels corresponding to the experiments with the different fractions of modified actin (former Figure 8A and C corresponding to the new Figure 8D and F).

Reviewer 2 Report

Minor comment:

The manuscript mentions a large number of actin-binding proteins, whose names are abbreviated. It is not easy to remember these abbreviations while  reading the text. Therefore the text needs to be supplemented with a table containing the decoding of the mentioned proteins/

Author Response

Please find our point-by-point response to the reviewer’s comments below in green.

Reviewer #2

 The manuscript by Songyu Dong and coauthors “Photorhabdus luminescens TccC3 toxin targets the dynamic population of F-actin and impairs cell cortex integrity” is well characterized by its title. The TccC3 is an effector domain of the ABC-toxin produced by entomopathogenic bacteria Photorhabdus luminescens that ADP-ribosylates actin at Thr148. As Thr148 is located at the site involved in the interaction with numerous G- and F-actin binding proteins, the effects of the ADP-ribosylated G- and F-actin on their interaction with actin-binding proteins were studied. The results of the work show that tropomyosin protects F-actin from the ADP-ribosylation, and only actin unprotected by tropomyosin is the physiological TccC3 substrate.

It is also shown that the fully modified F-actin is resistant to depolymerization by cofilin, actin depolymerizing factor and gelsolin, while ADP-ribosylation of G-actin impedes the ability of thymosin-beta 4 and profilin to inhibit spontaneous nucleation of actin filaments. Similar experiments were performed with the toxin-modified G-actin. Together, these experiments showed that while actin Thr148 is located at a site involved in interactions with numerous G- and F-actin-binding proteins, only a fraction of actin can be accessible for the enzyme binding. This allowed the authors to suggest that the TccC3 toxin works in those parts of eukaryotic cells that are characterized by high dynamics of the actin cytoskeleton. Upon injection to the cytoplasm of host cells, TccC3 attacks the fraction of F-actin that is not protected by tropomyosin. Since thymosin- beta 4 and profilin-bound G-actin also cannot be utilized as substrates for TccC3, the activity of the toxin can be shifted towards a subpopulation of mostly branched lamellipodial F-actin networks or a G-actin pool directly involved in such activities as cell migration, endo- and phagocytosis, membrane and organelle remodeling. This interpretation of the results allowed the authors to connect the properties and interactions of contractile proteins in vitro with the dynamics of the membrane-bound actin cytoskeleton in vivo, which makes their work particularly interesting.

Minor point to be addressed:

The article mentions a large number of actin-binding proteins, whose names are abbreviated. Remembering these abbreviations is not easy, therefore the article needs to be supplemented with a table containing the decoding of the mentioned proteins.

The manuscript can be certainly recommended for publication.

We thank the reviewer for the careful reading of the manuscript and providing the encouraging evaluation. We have followed the suggestion and added new Table 1 that summarizes full names and abbreviations of the proteins used in the study. 

Round 2

Reviewer 1 Report

I appreciate the efforts made by the authors to address the revisions. The requested structural models have been added. Some experiments have been done or attempted. The manuscript has been corrected appropriately.

Minor point to be adressed :

To explain the high variation, in different cells, of actin, PFN1, and TMSB4 concentrations, following references should be added (page 9).

Pollard T. D., Blanchoin L. and Mullins R. D., Molecular mechanisms controlling actin filament dynamics in nonmuscle cells, Annual Review of Biophysics and Biomolecular Structure, 2000

Funk J., Merino F., Venkova L., Heydenreich L., Kierfeld J., Vargas P., Raunser S., Piel M., Bieling P., Profilin and formin constitute a pacemaker system for robust actin filament growth, eLife, 2019

I believe this manuscript has been sufficiently improved to warrant publication in IJMS.

Author Response

Thank you for the valuable suggestion. We followed the advice and included the two references on the page 9 (new refs #29 and 30).